# Recurrence Patterns and Risk Factors after Curative Resection for Colorectal Cancer: Insights for Postoperative Surveillance Strategies

**DOI:** 10.3390/cancers15245791

**Published:** 2023-12-10

**Authors:** Hyo Seon Ryu, Jin Kim, Ye Ryung Park, Eun Hae Cho, Jeong Min Choo, Ji-Seon Kim, Se-Jin Baek, Jung-Myun Kwak

**Affiliations:** Division of Colon and Rectal Surgery, Department of Surgery, Korea University College of Medicine, 73 Goryeodae-ro, Sungbuk-gu, Seoul 02841, Republic of Korea; dr_hyo@korea.ac.kr (H.S.R.); jskpyr1224@naver.com (Y.R.P.); cate.eunhae.cho@gmail.com (E.H.C.); jmc2199@naver.com (J.M.C.); weehope621@naver.com (J.-S.K.); xezin@korea.ac.kr (S.-J.B.); jmkwak@korea.ac.kr (J.-M.K.)

**Keywords:** colorectal cancer, recurrence, risk factors, surveillance, liver metastasis, lung metastasis, peritoneal metastasis

## Abstract

**Simple Summary:**

In this retrospective study, we investigated the recurrence patterns and associated risk factors following curative resection. Peak recurrence occurred at 11 months, showing varied patterns based on tumor location, stage, and risk factors, such as vascular or perineural invasion, anastomotic leakage and circumferential resection margin involvement. Short-interval surveillance within the first 2 years after surgery, especially for high-risk patients in the early period, is warranted.

**Abstract:**

This study aimed to assess recurrence patterns and related risk factors following curative resection of colorectal cancer (CRC). This retrospective observational study was conducted at a tertiary care center, including 2622 patients with stage I–III CRC who underwent curative resection between 2008 and 2018. Hazard rates of recurrence were calculated using a hazard function. The primary outcome was the peak recurrence time after curative resection and secondary outcomes were prognostic factors associated with recurrence. Over a median follow-up period of 53 months, the overall, locoregional and systemic recurrence rates were 8.9%, 0.7%, and 8.5%, respectively. Recurrence rates were significantly higher for rectal cancer (14.9% overall, 4.4% locoregionally, and 12.3% systemically) than for colon cancer (all *p* < 0.001). The peak recurrence time was 11 months, with variations in hazard rates and curves depending on the tumor location, stage, and risk factors. Patients with AL or CRM involvement exhibited a distinct pattern, with a high hazard rate in the early postoperative period. Understanding these recurrence patterns and risk factors is crucial for establishing effective postoperative surveillance strategies. Our findings suggested that short-interval surveillance should be considered during the first 2 years post-surgery, particularly for high-risk patients who should receive early attention.

## 1. Introduction

The incidence rate of colorectal cancer has shown a declining trend globally; however, it still maintains high occurrence and mortality rates. Advancements in early detection through national screening programs, improvements in surgical techniques, neoadjuvant treatments, and the development of new anticancer drugs have significantly improved the treatment outcomes for colorectal cancer [1,2]. The peak mortality rate has decreased by over 50% compared to that in the 1980s [3,4]. Despite these improvements, the recurrence rate after curative resection of colorectal cancer is approximately 30% [5]. 

Postoperative surveillance for patients with colorectal cancer following curative resection aims to monitor postoperative complications and detect recurrence and metachronous cancers at their early stages of treatment. According to the literature, approximately 80% of all recurrences manifest within the initial 3 years after surgery, with 95% occurring within 5 years [6,7]. Consequently, most clinical guidelines consistently recommend a 5-year surveillance period [8,9,10,11,12,13]. Typically, surveillance protocols include recommendations for clinical assessment and carcinoembryonic antigen (CEA) tests every 3–6 months and abdominopelvic computed tomography (APCT) and chest CT scans every 6–12 months, particularly during the first 3 years. Recommendations for colonoscopy vary slightly, typically suggesting it to be performed 1–3 times within a 5-year period. However, the lack of consensus regarding surveillance intervals and diagnostic modalities has resulted in variations in surveillance policies across different guidelines, institutions, and individual healthcare providers. 

Despite being categorized as colorectal cancers, colon and rectal cancers manifest notable differences in tumor behavior, metastatic pathways, and recurrence patterns, which are attributed to distinct embryological and anatomical features. Additionally, the unique anatomical challenges posed by rectal cancer surgery and recent adoption of multidisciplinary approaches have brought about significant changes in treatment strategies for rectal cancer. Recent guidelines have shown a trend in distinguishing between surveillance strategies for colon and rectal cancers. The National Comprehensive Cancer Network guidelines recommend proctoscopy with transrectal ultrasonography or colonoscopy for patients with rectal cancer who have undergone local excision only or are in stage I [9]. The European Society for Medical Oncology briefly mentions the potential need for active surveillance of local recurrence in patients who are at a high risk of developing rectal cancer, such as those with circumferential resection margins (CRM) [10]. The Japanese Society for Cancer of the Colon and Rectum includes an additional colonoscopy in the second year of surveillance for rectal cancer [12].

Therefore, a comprehensive understanding of the recurrence patterns based on the primary site and risk factors, including not only the stage, but also the clinical and treatment-related factors associated with recurrence, is crucial for establishing optimal postoperative surveillance strategies. This study aimed to assess recurrence patterns and risk factors associated with recurrence after curative resection of colon and rectal cancers.

## 2. Materials and Methods

### 2.1. Study Population

This retrospective study was conducted at a tertiary hospital in Korea. The medical records of 3645 patients with colorectal cancer diagnosed between January 2008 and December 2018 were obtained from the institutional database. To conduct an analysis of the results from a 5-year longitudinal observation, only patients up to December 2018 were included. Patients with pathological stages I–III colorectal cancer who underwent curative resection were considered eligible for inclusion. We excluded patients with initial stage IV, inflammatory bowel disease-associated colorectal cancer, synchronous or metachronous colorectal cancer, hereditary colorectal cancer, carcinoma in situ, and those who underwent only local excision. The final study cohort comprised 1591 and 1031 patients with colon cancer and rectal cancer, respectively (Figure 1). The following parameters were retrospectively assessed through medical records: age, sex, tumor location, pathologic findings, preoperative obstruction or perforation, surgical information, neoadjuvant or adjuvant treatment, and recurrence. Cancer staging was determined based on the American Joint Committee on Cancer manual at the time of surgery. The study population was divided into a colon cancer group and a rectal cancer group for comparative analysis. The primary outcome was the peak recurrence time after curative resection and secondary outcomes were prognostic factors associated with recurrence. 

This study was approved by the Institutional Review Board (IRB) of Korea University Anam Hospital (registration no. 2023AN0392, 9 September 2023), and the study protocol conformed to the tenets of the Declaration of Helsinki. The requirement for informed consent was waived by the IRB owing to the retrospective nature of the study.

### 2.2. Treatment and Postoperative Surveillance

Curative-intent surgery was performed according to the principles of complete mesocolic excision and central vessel ligation for colon cancer and total mesorectal excision for rectal cancer, as previously described [14,15]. All operations were performed by four experienced colorectal surgeons (50 colorectal cancer operations per year for >5 years) using standardized techniques. Surgery was usually performed laparoscopically, and in some cases, robotic or open surgery was also conducted. Adjuvant chemotherapy was recommended according to clinical guidelines, except for patients with serious comorbidities or poor general performance. The regimens and cycles of adjuvant chemotherapy were administered according to the standard protocol [8,9]. Neoadjuvant treatment for low to mid rectal cancer included external beam radiation at a total dose of 25 Gy in 5 Gy fractions or 50 Gy in 25 Gy fractions and concurrent oral capecitabine [9].

The patients underwent regular examination every 3 months for the first 2 years after surgery. History taking, physical examinations, and laboratory tests, including serum CEA levels, were performed every 3 months for 2 years and every 6 months thereafter. Chest CTs and APCTs were repeated every 6 months during the follow-up period. The colonoscopy was performed within 1 year of surgery and once every 2–3 years thereafter. If recurrence was suspected, additional imaging tests, such as pelvic or liver magnetic resonance imaging or positron emission tomography–CT were performed as needed.

### 2.3. Statistical Anayses

Quantitative variables were expressed as means with standard deviations when they followed a normal distribution, according to the Kolmogorov–Smirnov test results. Asymmetrically distributed variables were expressed as median values with interquartile range. Categorical variables were presented as numbers and frequencies. We used the chi-square test to compare the distribution of categorical variables and a *t*-test for continuous variables. Disease-free survival (DFS) and overall survival (OS) were calculated using the Kaplan–Meier method and compared using the log-rank test. Univariate and multivariate analyses were conducted using the Cox proportional hazards model to analyze hazard ratios (HRs), from which 95% confidence intervals (CIs) were obtained. Confounding factors were selected in a forward selection procedure with a limit of 5% change in the effect size, using a basic logistic regression model. The hazard function was estimated for recurrence, stratified by stage and risk factors. Smoothing was performed using a kernel function method [16]. The units of the hazard rate measures were events per month. Statistical significance was established using a two-sided test with *p* < 0.05. All statistical analyses were performed using IBM SPSS (version 25.0; IBM, Armonk, NY, USA) and R version 4.1.2 (R project, “bshazard” package).

## 3. Results

### 3.1. Patient Characteristics

Table 1 summarizes the clinical and tumor characteristics of the patients based on tumor location. Compared to patients with colon cancer, those with rectal cancer exhibited a higher proportion of male individuals (57.7% vs. 65.9%, respectively; *p* < 0.001). Among patients with rectal cancer, 30.2% underwent neoadjuvant chemoradiotherapy and 53% received adjuvant chemotherapy. Patients with colon cancer were significantly older than patients with rectal cancer (64.2 vs. 61.6, respectively; *p* < 0.001). The presence of obstruction was significantly higher (12.5% vs. 3.1%, respectively; *p* < 0.001) and an endoscopic stent insertion was performed more frequently (5.2% vs. 1.3%, respectively; *p* < 0.001) in patients with colon cancer than in those with rectal cancer. Additionally, perforation was more commonly observed in patients with colon cancer than among those with rectal cancer (3.0% vs. 0.7%, respectively; *p* < 0.001). Among the patients with colon cancer, 48.4% received adjuvant chemotherapy. The characteristics based on the recurrence status in colon cancer and rectal cancer are presented in Appendix A, respectively. The distribution of neoadjuvant and adjuvant treatment by stage for patients with colon and rectal cancer is presented in Appendix A.

Table 2 presents the surgical and pathological characteristics of the patients based on tumor location. The rates of emergency surgery (2.7% vs. 0.1%, respectively; *p* < 0.001) and open conversion (6.2% vs. 1.6%, respectively; *p* < 0.001) were higher in colon cancer surgeries. Anastomotic leakage (AL) occurred more frequently in patients with rectal cancer than in those with colon cancer (10.2% vs. 3.1%, respectively; *p* < 0.001). Patients with colon cancer exhibited a higher tendency for T stage (*p* < 0.001), and lymphatic invasion was more common (17.3 vs. 13.3%, *p* = 0.028). For TNM stage classification, 23%, 40%, and 37% of patients with colon cancer had stage I, II, and III, respectively, and 1.9%, 30%, 32%, and 36% of patients with rectal cancer patients had, stage 0 (ypStage), I, II, and III, respectively (*p* = 0.002). Resection margin involvement (distal margin resection [DRM]: 0.7% vs. 0.1%, respectively; *p* = 0.005; CRM: 2.7% vs. 0.9%, respectively; *p* < 0.001) was more common in patients with rectal cancer than in those with colon cancer.

### 3.2. Survival Outcomes

The median follow-up duration was 53 months (interquartile range, 35–74 months). For patients with colon cancer, the 5-year DFS was 81.1% and 5-year OS was 85.1%. Additionally, for patients with stage I, II, and III colon cancer, the 5-year DFS rates were 92.6%, 81.9%, and 73.5%, respectively, and the 5-year OS rates were 94.2%, 85.0%, and 79.9%, respectively (*p* < 0.001). For patients with rectal cancer, the 5-year DFS was 75.9%, and the 5-year OS was 84.1%, respectively. Additionally, for patients with (yp) stages I, II, and III rectal cancer, the 5-year DFS rates were 87.8%, 76.3%, and 64.2% (*p* < 0.001), and the 5-year OS rates were 91.5%, 84.2%, and 77.4%, respectively (*p* < 0.001).

### 3.3. Recurrence Patterns and Risk Factors Associated with Recurrence

Table 3 summarizes the recurrence patterns of colon and rectal cancers after curative resection. Compared to patients with rectal cancer, those with colon cancer showed a relatively shorter median time to recurrence (15 vs. 14 months, respectively; *p* = 0.049). In our study population, the overall recurrence rates were 14.9%, 4.4%, and 12.3% for rectal cancer and locoregional and systemic recurrence, respectively. In 1.7% of the patients, both locoregional and systemic recurrences occurred simultaneously. Among patients with colon cancer, the overall recurrence rate was 8.9%, with locoregional recurrence at 0.7% and systemic recurrence at 8.5%. The incidence of combined locoregional and systemic recurrence was 0.4%. All recurrence rates were significantly higher for rectal cancer (*p* < 0.05). When stratified based on the site of metastasis, the involvement of the liver, lung, and distant lymph node (LN) metastases was significantly more prevalent among patients with rectal cancer than among those with colon cancer, whereas no statistically significant differences were noted in other metastatic sites. 

Univariate and multivariate analyses of the risk factors associated with recurrence were performed separately for each recurrence site (Table 4 and Table 5). The pathologic T stage was identified as an independent risk factor with a significantly high risk of all recurrences, while the N stage was observed to be associated with the risk of all recurrences, except for liver metastasis. The factors significantly associated with locoregional recurrence were rectal cancer (HR = 4.31; 95% CI = 2.08–8.94; *p* < 0.001), AL (HR = 2.86; 95% CI = 1.45–5.62; *p* = 0.002), CRM positivity (HR = 2.83; 95% CI = 1.01–7.94; *p* = 0.048), and neoadjuvant treatment (HR = 2.56; 95% CI = 1.42–4.61; *p* = 0.002). Perineural invasion (HR = 2.49; 95% CI = 1.28–4.86; *p* = 0.007) was a risk factor associated with liver metastasis, whereas rectal cancer (HR = 1.93; 95% CI = 1.20–3.13; *p* = 0.007), stent insertion (HR = 2.47; 95% CI = 1.11–5.49; *p* = 0.027), vascular invasion (HR = 2.29; 95% CI = 1.11–4.76; *p* = 0.026), and neoadjuvant treatment (HR = 2.58; 95% CI = 1.55–4.30; *p* < 0.001) were associated with a high risk of lung metastasis. Rectal cancer (HR = 1.92; 95% CI = 1.01–3.63; *p* = 0.046) and vascular invasion (HR = 3.64; 95% CI = 1.53–8.67; *p* = 0.004) were associated with the risk of distant LN metastasis. Poor differentiation was associated with peritoneal seeding (HR = 2.77; 95% CI = 1.39–5.51; *p* = 0.004) (Table 5). 

When performing multivariate analysis exclusively among patients with colon cancer, in addition to the T and N stages and vascular invasion, primary tumor obstruction was significantly associated with DFS (Appendix A). In patients with rectal cancer, in addition to the T and N stages, an LN harvest of less than 12, DRM and CRM involvement, clinical factors including primary tumor perforation, emergency surgery, intraoperative blood loss exceeding 800 mL, AL, and treatment factors including stent insertion and neoadjuvant treatment were identified as risk factors associated with recurrence (Appendix A).

### 3.4. Hazard Functions for Recurrence

The hazard function analysis enabled the visualization of the dynamics of recurrence over time. In patients with both colon and rectal cancer, the hazard function curves of recurrence peaked at 11 months and gradually decreased with a long slope to the right, without exhibiting differences based on the stage of cancer (Figure 2a,b). In patients with colon cancer, a relatively little difference was observed in the peak rates based on cancer stage. However, in rectal cancer, a noticeable increase in peak rates was observed as the stage advanced (Figure 2a,b). The hazard function curves for vascular and perineural invasions also displayed a peak at 11 months, with patients having vascular or perineural invasion exhibiting a high peak rate (Figure 3a,b). When considering locoregional and systemic recurrence separately, the hazard rate for systemic recurrence was reported to be higher for colon cancer than for rectal cancer at 11 months; however, the hazard rate curve subsequently decreased after the peak. Locoregional recurrence rates remained consistently high in patients with rectal cancer than in those with colon cancer, with a peak rate at 11 months, which was 3.7 times higher. In colon cancer, the hazard function curve for locoregional recurrence exhibited multiple peaks within 30 months, with a later peak at 20 months, and the peak rates were low (<0.00056) (Figure 4). Notably, in patients with AL, both locoregional and systemic recurrences displayed linear curves characterized by an early peak rate, followed by a gradual decrease in the slope. Moreover, patients with AL exhibited a locoregional recurrence peak rate that was 5.6 times higher than that of patients without AL, (Figure 5a). In patients with CRM involvement, a curve with an early peak rate, followed by a gradual decrease in the slope, was observed. Compared to patients without CRM involvement, those with CRM involvement exhibited a 3.6-fold higher hazard rate for locoregional recurrence and a 2.6-fold higher hazard rate for systemic recurrence (Figure 5b).

## 4. Discussion

In this study, we investigated the risk factors associated with recurrence based on tumor location and recurrence site and analyzed the time trend of recurrence in patients with colorectal cancer patients who underwent curative resection. The peak recurrence time was observed at 11 months after surgery, regardless of tumor location and stage. However, in rectal cancer, the hazard rate for systemic recurrence remained elevated for up to 2 years after surgery, followed by a gradual decline over time. The locoregional recurrence followed a similar pattern of recurrence to that of systemic recurrence, suggesting the necessity for intensive surveillance, including APCT, chest CT, and sigmoidoscopy, within the first 2 years after rectal cancer surgery. Conversely, for colon cancer, a steep increase was observed in the hazard rate for systemic recurrence at 11 months, followed by a rapid decline after approximately 15 months, whereas the risk of locoregional recurrence was lower, peaking at approximately 20 months. Therefore, imaging studies conducted approximately 1 year after surgery are crucial for detecting systemic recurrence rather than locoregional recurrence in patients with colon cancer. We highlighted that high-risk patients with AL or CRM involvement exhibited significantly elevated peak rates for both systemic and locoregional recurrences in the early period. Thus, short-interval surveillance is warranted for high-risk patients, starting immediately after surgery during the early period. 

In a large cohort study conducted by Kudose et al., peak recurrence times of 19, 13, and 11 months were observed for stages I, II, and III, respectively, with no significant variation based on tumor stage or location. Additionally, patients who underwent adjuvant chemotherapy exhibited a lower peak rate and a delayed peak month of 15.6 than those of patients who did not receive adjuvant chemotherapy [17]. Another nationwide study conducted in Japan, which focused on the timing of metastasis occurrence by organ, reported peak months of 11, 8, and 6 months for stages I, II, and III, respectively, which is earlier than that observed in our study. The peak time for liver metastasis was reported to be 8 months [18]. Notably, this Japanese study lacked information on neoadjuvant or adjuvant chemotherapy, which could potentially explain the earlier peak recurrence time and higher hazard rate observed in stages II and III than those observed in our study population.

Our findings suggested that increased attention should be paid to patients with AL or CRM involvement in the early stages. AL was predominantly observed in rectal cancer than in colon cancer. Patients with AL exhibited an early peak in the hazard rate of recurrence because of their higher risk of recurrence during the early stages. As time progressed, patients who experienced recurrence were excluded, resulting in a graph showing a decreasing hazard rate with a long hem to the right. The impact of AL remains debatable; however, numerous studies have demonstrated an association between AL and both local recurrence and survival outcomes [19,20]. Several theories have been proposed to explain the local recurrence in the context of AL, including an implantation of exfoliated tumor cells at the anastomotic site [21], metachronous carcinogenesis [21,22], and inflammation-mediated carcinogenesis [23]. In our study, AL was identified as a significant risk factor for locoregional recurrence, and patients who experienced AL exhibited a notably higher peak hazard rate in the early postoperative period than that of patients without AL. CRM is a significant prognostic factor for locoregional recurrence, as well as distant metastasis and overall survival outcomes. Its importance is more pronounced in patients who undergo neoadjuvant chemoradiotherapy. Biologically unfavorable tumors that survive radiotherapy result in CRM. Recent meta-analyses have shown that the relationship between CRM positivity and oncological outcomes is similar in patients who received neoadjuvant chemoradiotherapy and those who did not [24]. 

Our findings regarding recurrence sites and associated risk factors generally align with those of previous studies. The T and N categories were the most critical prognostic factors, and our study confirmed their association with all recurrences. Vascular and perineural invasions are markers for a more aggressive tumor phenotype and poor prognosis [25,26,27,28]. Some studies have reported an association between mucinous and signet ring cell differentiation and peritoneal dissemination [29]. Rectal cancer is associated with a higher incidence of lung metastasis and locoregional recurrence than colon cancer, consistent with its anatomical characteristics. [30,31]. The impact of stent insertion on oncologic outcomes has consistently yielded inconclusive results in previous studies [32,33,34,35]; however, a meta-analysis involving eight randomized controlled trials revealed a significant association with a high recurrence rate [36]. Theoretically, the enforced radical dilatation caused by stent placement may manipulate the tumor, potentially leading to the spread of cancer cells into the surrounding lymphatic vessels or peripheral bloodstream. Yamashita et al. demonstrated an increase in viable circulating tumor cells in the blood following stent insertion [37]. In the treatment of rectal cancer, the introduction of neoadjuvant chemoradiotherapy has led to a significant reduction in local recurrence rates [38]. In our study, the higher hazard ratio observed for locoregional recurrence and lung metastasis associated with neoadjuvant chemoradiotherapy was likely attributable to the fact that the patients who received neoadjuvant chemoradiotherapy predominantly had locally advanced rectal cancer. There is controversy regarding the role of adjuvant chemotherapy in rectal cancer patients who have undergone neoadjuvant treatment. However, it is generally acknowledged that adjuvant chemotherapy in high-risk stage II or III colorectal cancer plays a crucial role in lowering the recurrence rate, thereby enhancing DFS and demonstrating an association with OS [7]. Our analysis results may suggest that adjuvant chemotherapy does not appear to influence recurrence. However, it is difficult to conclude the effectiveness of adjuvant chemotherapy based on these results alone as multiple variables, such as regimen, cycles, and adherence, need to be considered when assessing the influence of adjuvant chemotherapy. Another consideration is that the proportion of our study cohort receiving adjuvant chemotherapy is relatively high, with about 64% in stage II and 85% in both colon and rectal cancers at stage III, which is notably higher compared to what has been reported in other cohort studies (Appendix A) [39,40]. The limited inclusion of patients not undergoing adjuvant chemotherapy may render the assessment of the impact of adjuvant chemotherapy less appropriate. 

Based on this information, the next challenge is to implement intensive surveillance to improve survival rates. Previous studies investigating surveillance programs and survival outcomes have reported that high-intensity surveillance does not lead to a reduction in mortality rates, indicating no observed survival advantages [5,41,42]. In the Follow-Up After Colorectal Surgery trial, a comparison between a group in which tests were performed only on symptomatic patients (the minimum group) and those who underwent regular CEA or CT scans revealed no significant differences in the overall recurrence detection rates. However, in the regular follow-up group, the rate of detecting treatable recurrence was three times higher, and a significantly greater number of patients received curative-intent surgical treatment for the detected recurrences [5]. With advancements in surgical treatments and anticancer drugs for recurrent disease, the treatment outcomes have been improving [43,44,45,46,47,48]. Consequently, early detection of treatable recurrences is likely to become crucial for improving survival outcomes in the future. 

Several limitations of this study should be considered when interpreting its results. First, this was a retrospective, single-center study, potentially introducing inherent selection bias and unknown confounding factors. Second, while we generally adhered to the surveillance protocol described in the Methods section, we came across instances where the surveillance intervals were longer or shorter owing to individual patient or clinical circumstances. Third, the impact of adjuvant chemotherapy could not be accurately assessed because of the lack of consideration of regimen, dose intensity, and adherence. Moreover, the pathological stages of patients with rectal cancer included those who underwent neoadjuvant chemoradiotherapy and experienced downstaging, making it challenging to exclude the influence of neoadjuvant chemoradiotherapy on stage-specific outcomes. Additionally, conducting an accurate assessment of the effects of neoadjuvant chemoradiotherapy is difficult. Nonetheless, this study, through hazard function analysis, dynamically depicted the recurrence pattern over time following colorectal cancer surgery and demonstrated the peak recurrence time. This temporal analysis provides an insight into the periods and timing during which intensive surveillance is crucial. Through an analysis of risk factors specific to the site of recurrence, we identified factors that deserve increased attention during surveillance, including not only tumor-specific factors, but also clinical and treatment-related factors. The findings of our study are expected to help establish appropriate surveillance intervals and modalities.

## 5. Conclusions

Our study findings revealed that after colorectal cancer surgery, the peak recurrence time was 11 months. Notably, recurrence patterns varied depending on tumor location and stage, leading to distinct locoregional and systemic recurrence patterns. High-risk patients with risk factors, such as AL or CRM involvement, require short-interval surveillance immediately after surgery to enhance early detection and improve survival outcomes.

## Figures and Tables

**Figure 1 cancers-15-05791-f001:**
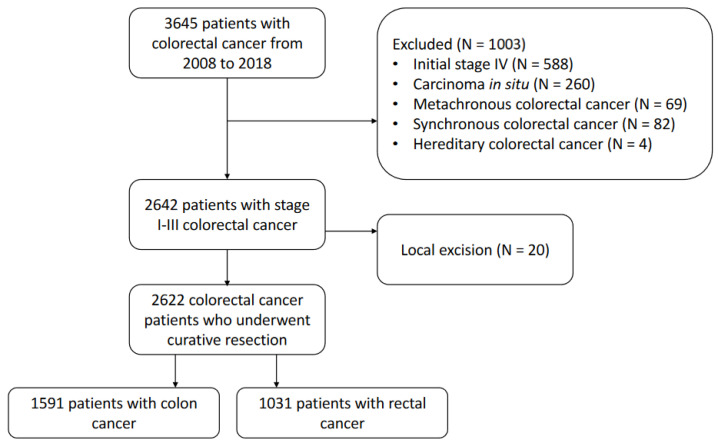
Flowchart of the study.

**Figure 2 cancers-15-05791-f002:**
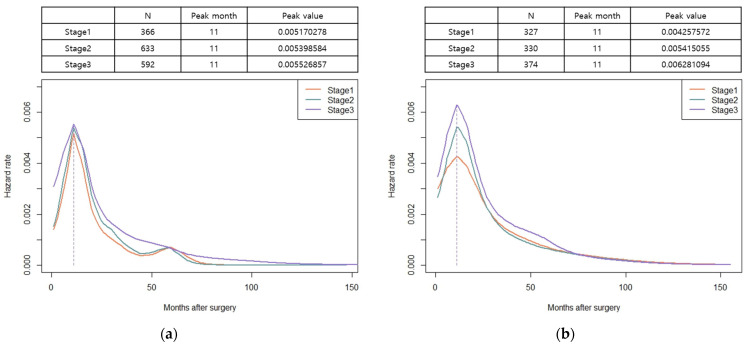
Hazard functions for recurrence stratified by stage, indicating conditional recurrence at time t. (**a**) Smoothed hazard functions for recurrence stratified by stage among patients with colon cancer; (**b**) smoothed hazard functions for recurrence stratified by stage among patients with rectal cancer. Units of measure for hazard rates were events per month.

**Figure 3 cancers-15-05791-f003:**
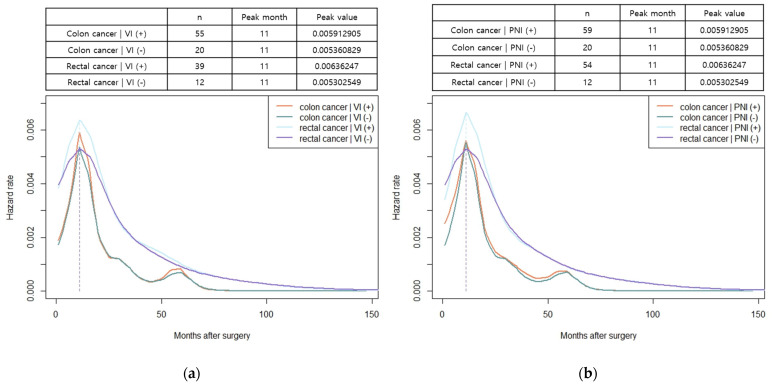
Hazard functions for recurrence stratified by vascular or perineural invasion, indicating conditional recurrence at time t. (**a**) Smoothed hazard functions for recurrence stratified by vascular invasion; (**b**) smoothed hazard functions for recurrence stratified by perineural invasion. Units of measure for hazard rates were events per month.

**Figure 4 cancers-15-05791-f004:**
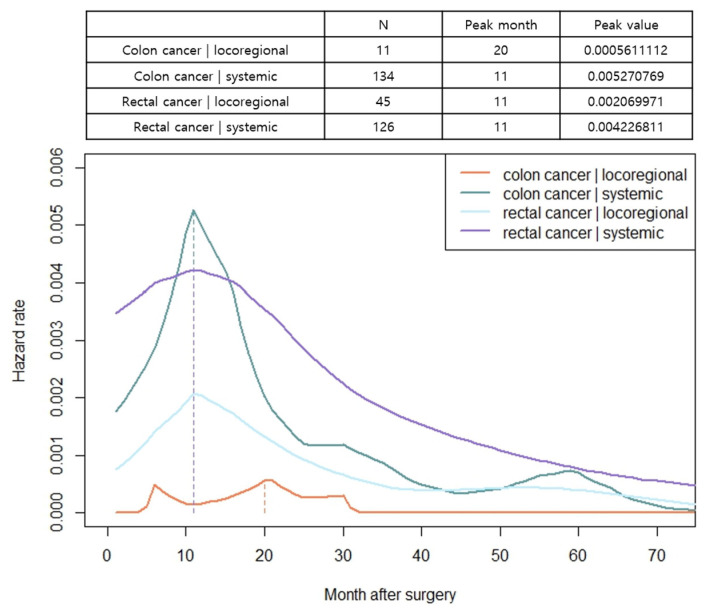
Hazard functions for locoregional and systemic recurrence stratified by tumor location, indicating conditional recurrence at time t. Units of measure for hazard rates were events per month.

**Figure 5 cancers-15-05791-f005:**
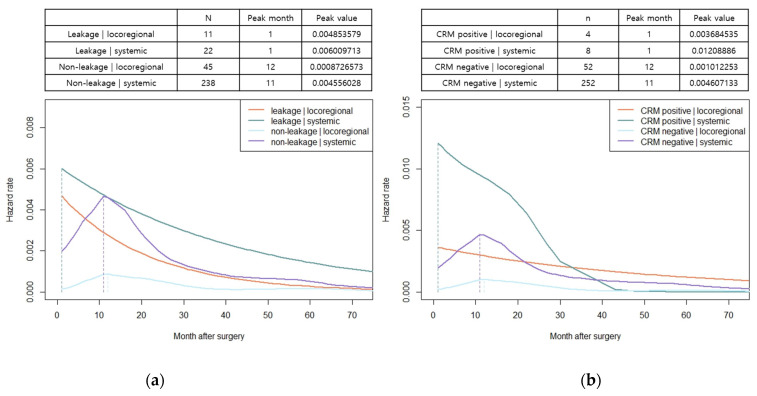
Hazard functions for recurrence stratified by recurrence site (locoregional and systemic), indicating conditional recurrence at time t. (**a**) Smoothed hazard functions for locoregional and systemic recurrence stratified by anastomotic leakage; (**b**) smoothed hazard functions for locoregional and systemic recurrence stratified by CRM involvement. Units of measure for hazard rates were events per month.

**Table 1 cancers-15-05791-t001:** Clinical and tumor characteristics based on tumor location.

	Colon (N = 1591)	Rectum (N = 1031)	*p* Value
Age, years (Mean ± SD)	64.2 ± 11.8	61.6 ± 12.1	<0.001
Sex, n (%)			<0.001
Male	918 (57.7)	679 (65.9)	
Female	673 (42.3)	352 (34.1)	
Body mass index, kg/m^2^ (Mean ± SD)	23.8 ± 3.3	23.6 ± 3.2	0.18
ASA score, n (%)			<0.001
1	296 (18.6)	240 (23.3)	
2	1155 (72.6)	733 (71.1)	
3	130 (8.2)	58 (5.6)	
4	10 (0.6)	0	
Comorbidities, n (%)			
Endocrine	342 (21.5)	205 (19.9)	0.32
Cardiovascular	64 (48.0)	419 (40.6)	<0.001
Respiratory	138 (8.7)	75 (7.3)	0.20
Smoking, n (%)			0.001
Yes	467 (29.4)	368 (35.7)	
No	1124 (70.6)	663 (64.3)	
Alcohol, n (%)			0.88
Yes	642 (40.4)	419 (40.6)	
No	949 (59.6)	612 (59.4)	
Tumor location, n (%)			
Ascending colon	410 (25.8)		
Transverse colon	261 (16.4)		
Descending colon	85 (5.3)		
Sigmoid colon	835 (52.5)		
Upper Rectum		151 (14.6)	
Mid/Low rectum		880 (85.4)	
CEA level ≥ 5 ng/mL, n (%)	321 (20.2)	189(18.3)	0.18
Obstruction, n (%)	199 (12.5)	32 (3.1)	<0.001
Perforation, n (%)	48 (3.0)	7 (0.7)	<0.001
Endoscopic stent insertion, n (%)	83 (5.2)	13 (1.3)	<0.001
Neoadjuvant therapy, n (%)	9 (0.6)	311 (30.2)	<0.001
Adjuvant chemotherapy, n (%)	770 (48.4)	546 (53.0)	0.021

ASA, American Society of Anesthesiologists physical status classification system; CEA, Carcinoembryonic antigen; SD, standard deviation.

**Table 2 cancers-15-05791-t002:** Surgical and pathological characteristics based on tumor location.

	Colon Cancer (N = 1591)	Rectal Cancer (N = 1031)	*p* Value
Emergency operation, n (%)	43 (2.7)	1 (0.1)	<0.001
Open/Laparoscopic/Robotic, n (%)	64 (4.0)/1496 (94.0)/31 (2.0)	10 (1.0)/524 (50.8)/497 (48.2)	0.02
Open conversion, n (%)	98 (6.2)	17 (1.6)	<0.001
R2 resection, n (%)	6 (0.4)	5 (0.5)	0.69
Estimated blood loss ≥ 800 mL	30 (1.9)	20 (1.9)	0.92
Operation time ≥ 240 min	224 (14.1)	497 (48.2)	<0.001
Transfusion, n (%)	94 (5.9)	62 (6.0)	0.91
Anastomotic leakage, n (%)	49 (3.1)	105 (10.2)	<0.001
Pathologic T stage, n (%)			<0.001
T0		24 (2.3)	
T1	234 (14.7)	129 (12.5)	
T2	212 (13.3)	253 (24.5)	
T3	1008 (63.4)	592 (57.4)	
T4	137 (8.6)	33 (3.2)	
Pathologic N stage, n (%)			0.64
N0	999 (62.8)	657 (63.7)	
N1	442 (27.8)	254 (24.6)	
N2	150 (9.4)	120 (11.6)	
TNM Stage, n (%)			0.002
0		20 (1.9)	
I	366 (23.0)	307 (29.8)	
II	633 (39.8)	330 (32.0)	
III	592 (37.2)	374 (36.3)	
Harvested LN < 12, n (%)	146 (9.2)	168 (16.3)	<0.001
Differentiation, n (%)			0.15
Well differentiated	302 (19.0)	194 (18.8)	
Moderately differentiated	1168 (73.4)	784 (76.0)	
Poorly differentiated	45 (2.8)	16 (1.6)	
Mucinous	70 (4.4)	34 (3.3)	
Signet ring cell	3 (0.2)	2 (0.2)	
Other types	3 (0.2)	1 (0.1)	
Lymphatic invasion, n (%)	275 (17.3)	137 (13.3)	0.028
Vascular invasion, n (%)	55 (3.5)	39 (3.8)	0.90
Perineural invasion, n (%)	59 (3.7)	54 (5.2)	0.26
DRM involvement, n (%)	1 (0.1)	7 (0.7)	0.005
CRM involvement, n (%)	15 (0.9)	28 (2.7)	<0.001

LN, Lymph node; DRM, Distal resection margin; CRM, Circumferential resection margin.

**Table 3 cancers-15-05791-t003:** Recurrence patterns of the two types of tumors.

	Colon Cancer (N = 1591)	Rectal Cancer (N = 1031)	*p* Value
Median time to recurrence (median, IQR)	14 (9–20.5)	15 (9–25.3)	0.049
Overall recurrence, n (%)	141 (8.9)	154 (14.9)	<0.001
Locoregional recurrence, n (%)	11 (0.7)	45 (4.4)	<0.001
Systemic recurrence, n (%)	136 (8.5)	127 (12.3)	<0.001
Combined locoregional and systemic recurrence, n (%)	6 (0.4)	18 (1.7)	0.008
Systemic recurrence site			
Liver	39 (2.5)	41 (4.0)	0.027
Lung	39 (2.5)	59 (5.7)	<0.001
Peritoneal seeding	36 (2.3)	23 (2.2)	0.96
Ovary	9 (0.6)	3 (0.3)	0.31
Distant LNs	21 (1.3)	27 (2.6)	0.015
Bone	5 (0.3)	7 (0.7)	0.18
^a^ Others	6 (0.4)	3 (0.3)	0.71

^a^ Others include the adrenal glands, spleen, brain, and vagina. IQR, interquartile range.

**Table 4 cancers-15-05791-t004:** Univariate analysis of risk factors associated with recurrence based on the recurrence site.

	Locoregional			Liver			Lung			Distant LNs			Peritoneal		
	HRs	95% CI	*p* Value	HRs	95% CI	*p* Value	HRs	95% CI	*p* Value	HRs	95% CI	*p* Value	HRs	95% CI	*p* Value
Rt colon vs. Lt colon	3.24	0.70–14.98	0.13	1.04	0.55–1.97	0.90	0.69	0.34–1.29	0.25	0.80	0.34–1.89	0.61	0.54	0.28–1.06	0.073
Colon vs. Rectum	6.37	3.29–12.31	<0.001	1.63	1.05–2.53	0.029	2.36	1.58–3.54	<0.001	1.20	1.13–3.54	0.017	1.02	0.60–1.73	0.94
Obstruction	1.39	0.60–3.24	0.45	2.02	1.09–3.73	0.025	1.62	0.88–2.96	0.12	1.67	0.71–3.93	0.24	3.32	1.79–6.16	<0.001
Perforation	1.91	0.47–7.82	0.37	1.32	0.33–5.38	0.70	1.07	0.26–4.34	0.93	1.11	0.15–8.02	0.92	0.90	0.12–6.49	0.92
Stent insertion	1.69	0.53–5.41	0.38	1.15	0.36–3.65	0.81	2.31	1.07–4.99	0.032	2.76	0.99–7.69	0.052	4.71	0.23–9.94	<0.001
Anastomotic leakage	4.48	2.32–8.66	<0.001	2.28	1.14–3.56	0.020	1.42	0.66–3.05	0.38	2.11	0.84–5.33	0.11	1.34	0.48–3.69	0.58
pT1/T2 vs. pT3/T4	2.66	1.31–5.43	0.007	6.24	2.72–14.34	<0.001	2.62	1.54–4.48	<0.001	3.55	1.51–8.36	0.004	4.39	1.89–10.23	0.001
N0 vs. N1/N2	2.29	1.35–3.88	0.002	2.34	1.52–3.67	<0.001	2.93	1.95–4.40	<0.001	2.58	1.45–4.58	0.001	2.80	1.66–4.84	<0.001
Differentiation	2.12	0.96–4.68	0.063	1.42	0.65–3.08	0.38	0.47	0.15–1.48	0.20	1.35	0.49–3.76	0.57	3.09	1.56–6.11	0.001
Vascular invasion	3.24	1.29–8.12	0.012	4.08	2.04–8.16	<0.001	3.40	1.71–6.74	<0.001	5.64	2.53–12.58	<0.001	3.05	1.22–7.63	0.017
Lymphatic invasion	0.82	0.37–1.80	0.62	2.02	1.23–3.32	0.006	1.12	0.66–1.92	0.67	1.91	0.995–3.68	0.052	1.64	0.89–3.04	0.12
Perineural invasion	2.71	1.08–6.81	0.033	5.29	2.92–9.60	<0.001	2.50	1.21–5.14	0.013	3.30	1.30–8.34	0.012	3.16	1.36–7.36	0.008
CRM	5.24	1.90–14.50	0.001	2.68	0.85–8.49	0.094	3.71	1.51–9.12	0.004	1.47	0.20–10.66	0.70	1.21	0.17–8.71	0.85
Neoadjuvant chemoradiotherapy	5.33	3.13–9.07	<0.001	2.05	1.20–3.51	0.008	3.26	2.11–5.03	<0.001	2.04	1.02–4.09	0.045	1.04	0.47–2.30	0.92
Adjuvant chemotherapy	2.14	1.21–3.78	0.009	1.54	0.98–2.42	0.063	2.33	1.50–3.61	<0.001	1.40	0.78–2.49	0.26	1.77	1.03–3.04	0.039

CRM, Circumferential resection margin; HR, hazard ratio; CI, confidence interval; Rt., right; Lt., left.

**Table 5 cancers-15-05791-t005:** Multivariate analysis of risk factors associated with recurrence based on recurrence site.

	Locoregional			Liver			Lung			Distant LNs			Peritoneal		
	HRs	95% CI	*p* Value	HRs	95% CI	*p* Value	HRs	95% CI	*p* Value	HRs	95% CI	*p* Value	HRs	95% CI	*p* Value
Rt colon vs. Lt colon															
Colon vs. Rectum	4.31	2.08–8.94	<0.001	1.54	0.91–2.59	0.11	1.93	1.20–3.13	0.007	1.92	1.01–3.63	0.046			
Obstruction				1.64	0.86–3.13	0.14							1.85	0.73–4.69	0.20
Perforation															
Stent insertion							2.47	1.11–5.49	0.027				1.96	0.64–5.99	0.24
Anastomotic leakage	2.86	1.45–5.62	0.002	1.91	0.93–3.89	0.076									
pT1/T2 vs. pT3/T4	2.56	1.18–5.55	0.017	4.69	2.00–11.01	<0.001	2.09	1.17–3.74	0.012	2.85	1.18–6.90	0.020	2.94	1.20–7.19	0.018
N0 vs. N1/N2	2.06	1.12–3.81	0.020	1.56	0.97–2.50	0.066	2.57	1.59–4.15	<0.001	1.91	1.05–3.48	0.034	2.49	1.34–4.66	0.004
Differentiation													2.77	1.39–5.51	0.004
Vascular invasion	2.50	0.93–6.74	0.071	1.81	0.82–3.97	0.14	2.29	1.11–4.76	0.026	3.64	1.53–8.67	0.004	1.76	0.65–4.82	0.27
Lymphatic invasion				1.38	0.80–2.40	0.25									
Perineural invasion	1.12	0.41–3.01	0.83	2.49	1.28–4.86	0.007	1.08	0.50–2.35	0.84	1.37	0.50–3.74	0.54	1.57	0.62–3.99	0.34
CRM	2.83	1.01–7.94	0.048				2.22	0.90–5.49	0.085						
Neoadjuvant chemoradiotherapy	2.56	1.42–4.61	0.002	1.77	0.95–3.29	0.073	2.58	1.55–4.30	<0.001	1.56	0.71–3.40	0.27			
Adjuvant chemotherapy	0.82	0.42–1.63	0.58				0.90	0.53–1.53	0.69				0.65	0.34–1.24	0.19

CRM: Circumferential resection margin; LNs, lymph nodes; HR, hazard ratio; CI, confidence interval; Rt., right; Lt., left.

## Data Availability

The data presented in this study are available on request from the corresponding author. The data are not publicly available due to privacy.

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
