# Peer review of "Recurrence Patterns and Risk Factors after Curative Resection for Colorectal Cancer: Insights for Postoperative Surveillance Strategies"

_cancers, 2023, doi:10.3390/cancers15245791_

Round 1
Reviewer 1 Report
Comments and Suggestions for Authors
In their single-center retrospective observational study, the authors examined recurrence patterns and risk factors associated with recurrence after curative resection of colon and rectal cancer (CRC) in 2,622 patients with stage I–III CRC who underwent curative resection between 2008 and 2018. They concluded that after colorectal cancer surgery, the peak recurrence was 11 months. They also pointed out that short-term surveillance within the first two years after surgery, especially for high-risk patients in the early period, is warranted.
I read the study with great interest. Unfortunately, important methodological issues were raised during the review. My concerns are as follows:
1. Abstract – The authors use the abbreviation CRC without explanation. Please provide the full title. The primary and secondary outcomes should be presented in the methodology of the abstract.
2. The authors state that the study was approved by the Institutional Review Board of Korea University Anam Hospital. Please include the date of approval, next to the approval number.
3. The methodology is inadequately presented. The detailed study design should be presented in the methodology. The authors should describe which parameters were recorded for each patient included in the study. In addition, the study groups should be clearly described in the methodology.
4. The authors should describe in the methodology which surgical procedures were used in the patients (description of procedures or appropriate references). They should indicate whether an open or laparoscopic approach was used for each procedure, as well.
5. The primary and secondary outcomes of this study should be presented in the methodology.
6. Why were all quantitative variables reported as means with standard deviations? What statistical test was performed to check the normality of the data distribution? The authors should specify a statistical test to check the normality of the data distribution. Only normally distributed data should be presented as mean (SD). Asymmetrically distributed data should be reported as median (IQR).
7. Authors should describe in the methodology why patients were not included in this analysis after December 2018.
8. The quality of the English should be improved. The manuscript should be edited by a native English speaker or a professional language editor to improve grammar and readability.
9. Conclusions gained from this study are not new, the authors should explain why this study is important and what is different compared to previously published data.
Comments on the Quality of English Language
The quality of the English should be improved. The manuscript should be edited by a native English speaker or a professional language editor to improve grammar and readability.
Reviewer 2 Report
Comments and Suggestions for Authors
Major points;
1) Figure 5. (a) and (b).
The peak time for detection of recurrence in Leakage (+) cases and CRM positive cases is 1 month, but this is before blood sampling or CT is performed according to the surveillances protocol. The number of these cases is quite small, but what kind of tests led to the discovery of recurrence? If recurrence was confirmed due to frequent CT scans for leakage confirmation and follow-up, the number of cases in which recurrence did not occur even after leakage and the clinical course of the cases should be shown. The same goes for CRM positive; unless authors show the testing method for detecting recurrence in cases that have recurred within three months and show the number of cases in which recurrence was not confirmed and the subsequent clinical course, it is not possible to suggest the need for early blood or CT check.
2) Tables 4. and 5.
The presence or absence of adjuvant chemotherapy did not contribute to recurrence, which is different from existing reports. The reason for this should be discussed not only in terms of dose intensity but also in terms of adjuvant chemotherapy indications and regimens.
3) Table 2.
There does not seem to be a significant difference in the values for Estimating blood loss, Transfusion, Pathologic N stage, and TNM stage with a p value of <0.001.
Reviewer 3 Report
Comments and Suggestions for Authors
The Authors reported recurrence patterns and risk factors after curative colorectal cancer resection and suggested postoperative surveillance strategies.
The manuscript has been well presented, written, and discussed.
Comments:
1. In this study, about 30% of rectal cancer patients received neoadjuvant treatment; however, their TNM staging has been classified pathologically (yTNM). TNM staging of this group of patients should be defined based on clinical and not pathological staging.
2. The authors should provide the distribution of neoadjuvant and adjuvant therapies among different stages of colon and rectal cancers in a separate table.
3. In table 5. the first row (Rt colon vs. Lt colon) is empty.
Round 2
Reviewer 1 Report
Comments and Suggestions for Authors
The authors responded appropriately to my comments and significantly improved the manuscript. I have no further comments or requests.
Comments on the Quality of English Language
The quality of English is slightly improved, but still requires some editing.
Reviewer 2 Report
Comments and Suggestions for Authors
Questions are well answered.
Reviewer 3 Report
Comments and Suggestions for Authors
Dear Authors,
Thank you for your revision.